# Light-Emitting Diodes Based on InGaN/GaN Nanowires on Microsphere-Lithography-Patterned Si Substrates

**DOI:** 10.3390/nano12121993

**Published:** 2022-06-10

**Authors:** Liliia Dvoretckaia, Vladislav Gridchin, Alexey Mozharov, Alina Maksimova, Anna Dragunova, Ivan Melnichenko, Dmitry Mitin, Alexandr Vinogradov, Ivan Mukhin, Georgy Cirlin

**Affiliations:** 1Department of Physics, Alferov University, Khlopina 8/3, 194021 St. Petersburg, Russia; liliyabutler@gmail.com (L.D.); gridchinvo@yandex.ru (V.G.); deer.blackgreen@yandex.ru (A.M.); george.cirlin@mail.ru (G.C.); 2Institute of Physics, Saint Petersburg State University, Universitetskaya Emb. 7/9, 199034 St. Petersburg, Russia; alex000090@gmail.com; 3Department of Physics, National Research University Higher School of Economics, Kantemirovskaya 3/1 A, 194100 St. Petersburg, Russia; anndra@list.ru (A.D.); imelnichenko@hse.ru (I.M.); 4Department of Chemistry, ITMO University, Lomonosova 9, 197101 St. Petersburg, Russia; mitindm@mail.ru (D.M.); avv@scamt-itmo.ru (A.V.); 5Higher School of Engineering Physics, Peter the Great St. Petersburg Polytechnic University, Polytechnicheskaya 29, 195251 St. Petersburg, Russia

**Keywords:** molecular beam epitaxy, nanowires, III-N, Si, microsphere lithography, light-emitting devices

## Abstract

The direct integration of epitaxial III-V and III-N heterostructures on Si substrates is a promising platform for the development of optoelectronic devices. Nanowires, due to their unique geometry, allow for the direct synthesis of semiconductor light-emitting diodes (LED) on crystalline lattice-mismatched Si wafers. Here, we present molecular beam epitaxy of regular arrays n-GaN/i-InGaN/p-GaN heterostructured nanowires and tripods on Si/SiO_2_ substrates prepatterned with the use of cost-effective and rapid microsphere optical lithography. This approach provides the selective-area synthesis of the ordered nanowire arrays on large-area Si substrates. We experimentally show that the n-GaN NWs/n-Si interface demonstrates rectifying behavior and the fabricated n-GaN/i-InGaN/p-GaN NWs-based LEDs have electroluminescence in the broad spectral range, with a maximum near 500 nm, which can be employed for multicolor or white light screen development.

## 1. Introduction

The direct growth of III-V and III-N nanostructures on Si substrates is one of the most promising means for the development of a new generation of optoelectronic devices [1,2,3]. Nanowires (NWs), having quasi-one-dimensional structures, are considered as building blocks for such devices, since these structures can be directly grown on lattice-mismatched Si substrates and possess high crystal perfection [4,5]. NWs demonstrate high crystal quality owing to the small footprint and effective mechanical stress relaxation on the developed side surface. Solid alloys of Ga(In, Al)N are often used for the fabrication of light-emitting and light-absorbing devices operating in a broad spectral range [6,7,8,9,10,11,12]. Light-emitting diodes (LEDs) based on InGaN/GaN heterostructured NWs have been successfully demonstrated, showing excellent performances in the blue spectral range [13,14,15,16]. NWs-based LEDs are considered to be the alternative to conventional organic-based solutions [17]. Moreover, the epitaxial growth of heterostructured III-N NWs with controllable doping profiles on relatively cheap Si substrates paves the way for the integration of III–V materials with an established complementary metal-oxide-semiconductor (CMOS) technology [18].

Using molecular beam epitaxy (MBE), Ga(In)N NWs can be directly synthesized on Si substrates in the form of self-induced disordered arrays [19,20,21]. Furthermore, within one epitaxial run, MBE enables the simultaneous growth of both vertically-aligned NWs and tripods [22] or even nanotube-like [23] structures, which can have different compositions of In. This can be employed for white light screen fabrication. Metalorganic chemical vapor deposition (MOCVD), similar to MBE, was employed to achieve the selective epitaxy of the arrays of heterostructured NWs [24,25,26]. MOCVD is considered to be a promising epitaxial technique for III-V and III-N mass production, allowing for the time-efficient synthesis of NWs-based heterostructures simultaneously on the set of large-area substrates. However, for the development of NWs-based applications, especially LEDs, the regular arrangement of nanostructures is required. Time-consuming approaches for direct lithography, such as e-beam lithography [27] or focused ion beam milling [28,29], are not fully applicable for the prepatterning of large-area substrates.

Among other approaches, microsphere optical lithography, which provides submicrometer-scale lateral resolution, is one of the most versatile, scalable and cost-effective methods for photoresist patterning [30]. Moreover, the design of the patterning can be easily tuned by the appropriate choice of the diameter of microspheres, while spin-coating enables the covering of the large-area substrates [31,32]. Selective-area epitaxy based on the Si/SiO_2_ growth substrates patterning with microsphere lithography allows for the obtention of the ordered arrays of NWs with a narrow distribution in geometrical sizes, which is essential for device processing [33,34].

In this work, we employ microsphere lithography for Si/SiO_2_ substrates preparation, allowing further selective-area MBE growth of the regular arrays of n-GaN/i-InGaN/p-GaN heterostructures. We show that the n-GaN NWs/Si substrate interface demonstrates rectifying electrical properties that are appropriate for LED fabrication. The produced NWs-based LEDs have the value of a knee voltage typical for III-N devices and show electroluminescence in a broad spectral range, which can be employed for multicolor or white light screen development.

## 2. Materials and Methods

### 2.1. Si/SiO_2_ Substrate Patterning

For the growth of heterostructured InGaN/GaN NWs, we employed MBE on prepatterned Si/SiO_2_ crystalline substrates. To estimate the electrical properties of the n-GaN/Si interface, we also synthesized an array of n-GaN NWs without InGaN active insertions and p-GaN shell layers on a Si prepatterned substrate. 

To make the NWs’ growth mask, Si substrates (n-doped to the level of 1 × 10^16^ cm^−3^) were thermally oxidized, that provided the formation of a 60 nm-thin layer of oxide. Then, we employed microsphere lithography and plasma etching to pattern the oxide layer in order to fabricate a growth mask for the selective-area epitaxy of the ordered arrays of the NWs. 

Microspheres were spin-coated on the layer of the photoresist covering a growth substrate and formed a dense monolayer array. The optimal parameters of microsphere deposition are presented in our previous work [32]. Then, we used ultraviolet (UV) flood exposure with a 365 nm wavelength to illuminate the photoresist. Every microsphere worked as a lens, focusing the UV light into the optical jet underneath [35]. During the development of the exposed photoresist, the spheres were spin off from the substrate; thus, the patterned resist layer served as a mask for the further inductively coupled SF6 etching of SiO_2_. Finally, the resist layer was removed, and we obtained the patterned Si/SiO_2_ substrates with arrays of the ordered submicron holes in the oxide layer. The workflow of the Si/SiO_2_ substrate patterning and the typical scanning electron microscopy (SEM) images of the fabricated substrates are presented in Figure 1. The developed approach allows for the patterning of large-area Si substrates from several cm^2^ up to wafers several inches in diameter.

We used microspheres that were 1.8 µm in diameter, which defined the period of the ordered arrays of the NW. This provided the elimination of possible issues, such as the competitive diffusion of the growth material over the substrate at nucleation and the initial stages of the NWs’ growth, as well as the shadowing of the NWs with a relatively small height. These effects can have a negative impact on the epitaxial synthesis of LED structures. 

### 2.2. MBE Growth

The MBE growth of NWs was carried out using Riber Compact 12 equipped with a nitrogen plasma source, providing the flux of nitrogen ions. Prior to the growth, the patterned Si/SiO_2_ substrates were heated up to a temperature of 915 °C and treated for 20 min, which enabled the removement of a thin native oxide layer. This process was controlled by in situ reflection high-energy electron diffraction. It should be noted that heating to this temperature enables native oxide desorption without the destruction of the growth oxide mask. After that, the substrate temperature was decreased to 830 °C, the nitrogen plasma source was ignited and the shutters of the Ga and Si cells were simultaneously opened. The nitrogen flow and the power of the nitrogen plasma source were 0.4 sccm and 450 W, respectively. The Ga beam equivalent pressure was equal to 3 × 10^−^^7^ Torr. By the end of the n-doped NW cores, the growth the shutter of the Si cell was closed to form an undoped part of GaN NWs with an estimated height of 15–20 nm. This eliminated the emergence of doping atoms in the active InGaN insertions. To grow the active InGaN insertions, the effusion cells were shut and substrate temperature was decreased to 660 °C. After that, the shutters of the Ga and In cells were opened. The Ga and In fluxes were held constant at 1 × 10^−7^ Torr. The estimated height of the grown active InGaN insertions was 30 nm. The growth of the p-type emitters was performed with a Mg effusion cell at the same temperature, which provided the formation of p-GaN shells, covering the whole length of the NWs. The thickness of the synthesized p-doped shells was estimated at 200 nm. Similar to the NW core emitters, the first 15–20 nm of the shells were grown without Mg doping. It should be noted that the formation of the emitter covering the entire nanowires was essentially important for the further post processing of the LED structure. 

Figure 2a,b show schematic views and typical SEM images of the arrays of n-GaN and n-GaN/i-InGaN/p-GaN NWs grown on n-Si substrates. According to the SEM images, the patterned SiO_2_ layers enabled the selective MBE growth of NWs in every hole of the mask. Note that the chosen MBE regime provided the nucleation and growth of both vertically aligned NWs and tripod nanostructures, which can be caused by an insufficiently high growth temperature or by the features of NW nucleation in the holes of the mask [36,37]. One can also note that the decreased temperature—required for the NW active area and shell formation—enabled the nucleation of a 2D parasitic layer on the surface of the SiO_2_ mask.

Another important peculiarity of the synthesized nanostructures is the changing of the NW facets during the growth from the NW base to the top (see the insert in Figure 2b). This can be governed by two factors: a rotation of the crystal lattice by 30 degrees or a change in the dominant facet. For more deep analysis, we performed an electron diffraction study near the base and the top of the NWs (see Appendix A for details). The acquired electron diffraction patterns are the same for both points, which proved the change in the dominant facet. One possible reason for this phenomenon is associated with the mechanical stress in the NWs that originated from the lattice mismatching between the GaN core and the InGaN active area. Another possible reason can be related to the features of Mg doping of the GaN shell.

## 3. Results and Discussion

### 3.1. Optical Properties Study

To evaluate the composition of active InGaN insertions, we performed a photoluminescence (PL) study. Figure 3 shows a typical PL spectrum obtained from the GaN/InGaN/GaN NW arrays. One can see that the InGaN insertions demonstrated a PL signal in a wide spectral range from visible to near infrared (IR), while the PL maximum is located near 500 nm. The relatively broad PL peak can be caused by the decomposition of the InGaN insertions on the phases with different contents of In [38] or by different In incorporations in non-polar and polar wurtzite planes [39]. It also should be mentioned that the inclined NWs and 2D parasitic layer can contribute to the PL signal, which can broaden the spectrum. 

### 3.2. Device Processing

Next, we carried out the postprocessing of the synthesized structures in order to fabricate the functionalized devices. The workflow of processing is shown in Figure 4. To fabricate ohmic contacts to Si substrates, their back sides were treated with 10% HF aqueous solution in order to remove the native oxide layer and passivate the surface with hydrogen. Immediately after that, the substrates were loaded into a vacuum chamber of a thermal evaporator BocEdwards Auto 500 to deposit Al contact with a thickness of 200 nm. Then, using spin-coating, the sides of the substrates with the NWs were covered with photo-curing epoxy resin (SU-8 negative photoresist). This provided the electrical isolation between the substrate and front contact. The thickness of the SU-8 layer was 100–200 nm less than the average length of the NWs. To remove the epoxy residue from the ends of the NWs, the substrates were treated in oxygen plasma. In the next technological step, we formed the mesa front contacts, which required the use of optical lithography, conductive layer deposition and the lift-off procedure. In the case of the test n-GaN NWs/n-Si structures, we evaporated the Al layer, which provided ohmic contact to GaN, while for the LED InGaN NWs-based structures, a thin layer of indium tin oxide (ITO) was magnetron sputtered. The ITO formed a transparent contact to the p-GaN shells of the LED structures. Note that only the longest NWs were in contact with the ITO, while the others were buried in the SU-8 layer.

### 3.3. Electrical and Electroluminescent Characterization

The current-voltage (I-V) characteristics of the fabricated devices were measured with the use of a Keithley 2401 source-meter. The samples were placed on a metallic table of a probe station with a vacuum clamp. A contact to the face electrode was organized using soft CuBe probes, which mitigated the mechanical scratching of the structures. Figure 5 presents the typical I-V curves for the test and LED structures. For all of the measurements, the positive input of the power supply was connected to the probe.

Figure 5a shows a typical I-V curve for one of the test devices, where the NWs contained only n-GaN cores (without InGaN active insertions and p-GaN shells). One can see that the curve demonstrates rectifying behavior. Moreover, as shown in the additional experiments (not presented here), for this type of device, the reverse bias current depends on the level of illumination with visible or IR light. Together with the polarity of the I-V curve, these evidence the emergence of a rectifying junction in the Si substrate. We suggest that this can be caused by a doping to the p-type conductivity of the Si surface layer by Ga atoms during MBE growth. The device demonstrates a current density up to 30 A/cm^2^ under an applied positive voltage of 2V and more than 100 A/cm^2^ under a voltage of 3V (see the positive branch of the I-V curve in Figure 5a). Thus, a drop in the voltage on the interface of n-GaN/n-Si should be taken into account while analyzing the I-V characteristic of LED structures.

Figure 5b presents a typical I-V curve for one of the LED n-GaN/i-InGaN/p-GaN devices. The knee voltage is about 6 V, while the typical current density corresponds to the level of several A/cm^2^. Considering the drop in the voltage on the n-GaN/n-Si interface discussed previously, we can conclude that, for this level of current density, around 1–1.5 V drops on the interface and the other voltage drops on the n-GaN/i-InGaN/p-GaN structure and the p-GaN/ITO interface appeared. Thus, the real knee voltage can be found to be around 4.5 V, which corresponds well to the expected value for this type of device. The electroluminescence (EL) of the LED devices can be detected by the naked eye for a current density exceeding 2 A/cm^2^. However, the estimation of the current density passing through the GaN/InGaN/GaN NWs is far from evident, since only part of the NWs is contacted to the ITO and, therefore, involved in the passing of current.

Figure 6 shows the obtained EL spectrum for the LED device. Note that only a small part of the NWs (the longest ones) was contacted by the ITO and, thus, could emit light. The EL spectrum demonstrates a dominant peak near 500–520 nm (corresponding to the cyan-green color on the optical images) and a wide tail near the IR spectral range. This result corresponds well to the experimentally measured PL spectra (see Figure 3). The difference in spectra (the drop in the EL signal in the range of 550–650 nm wavelengths) can be explained by the fact that only the longest (vertically aligned) NWs contribute to the measured EL signal. According to our experiments, the increase in current density passing through the device led to the growth of the IR tail on the EL spectra, which can be associated with the thermal heating of the NWs. However, the detailed analysis of this effect requires additional study.

## 4. Conclusions

In this work, we synthesized test n-GaN NWs and LED n-GaN/InGaN/p-GaN NWs using selective-area MBE growth on Si substrates with prepatterned SiO_2_ layers. To achieve the ordered arrays of the NWs, we employed microsphere lithography, providing the handling of the large-area substrates. The PL measurements revealed that the synthesized NWs have a relatively broad optical response, indicating the decomposition of the InGaN insertions on the phases with different contents of In.

We studied the transport properties of the n-GaN/n-Si interface and demonstrated rectifying behavior on the I-V curves. The knee voltage of the fabricated n-GaN i-InGaN/p-GaN structures was found to be around 4.5 V, which is typical for GaN-based LED devices. The measured EL spectra for the fabricated LED devices demonstrated a peak located around 500–520 nm and wide IR tail. The results of the EL measurements are consistent with the PL study.

## Figures and Tables

**Figure 1 nanomaterials-12-01993-f001:**
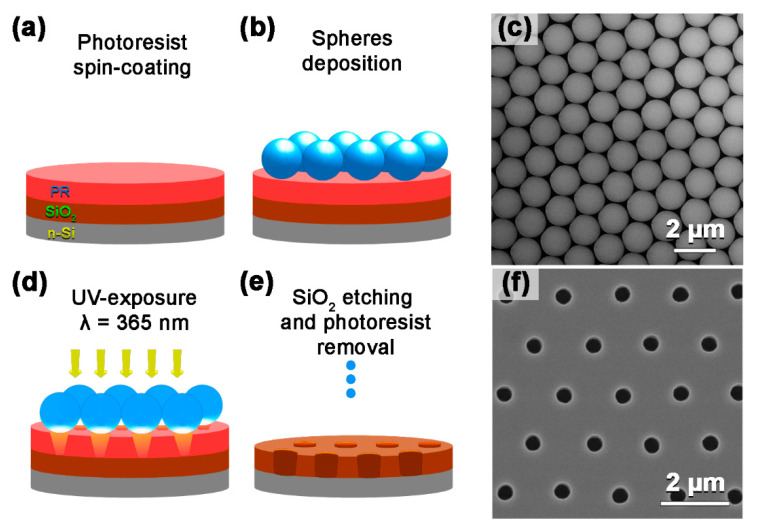
Workflow of Si/SiO_2_ substrate patterning: (**a**) spin-coating of the photoresist, (**b**) microsphere deposition, (**d**) UV exposure of the photoresist through a monolayer of microspheres, (**e**) SiO_2_ layer etching through the patterned photoresist. SEM images of the arrays of microspheres deposited on the photoresist layer (**c**) and microholes in the SiO_2_ mask on the Si substrate (**f**).

**Figure 2 nanomaterials-12-01993-f002:**
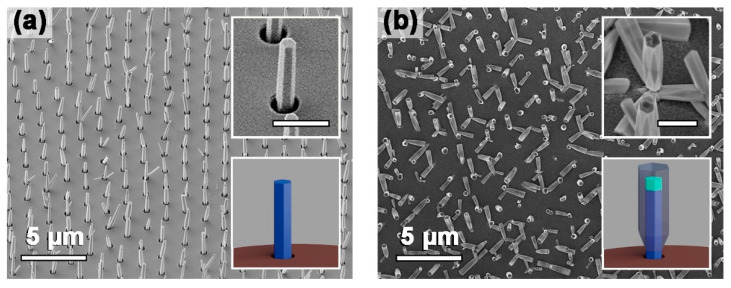
SEM images of the arrays of (**a**) n-GaN and (**b**) n-GaN/i-InGaN/p-GaN NWs grown on n-Si substrates. The inserts show the schematic view of the synthesized nanostructures (not in scale) and the enlarged SEM images (the scale bar is 1 µm).

**Figure 3 nanomaterials-12-01993-f003:**
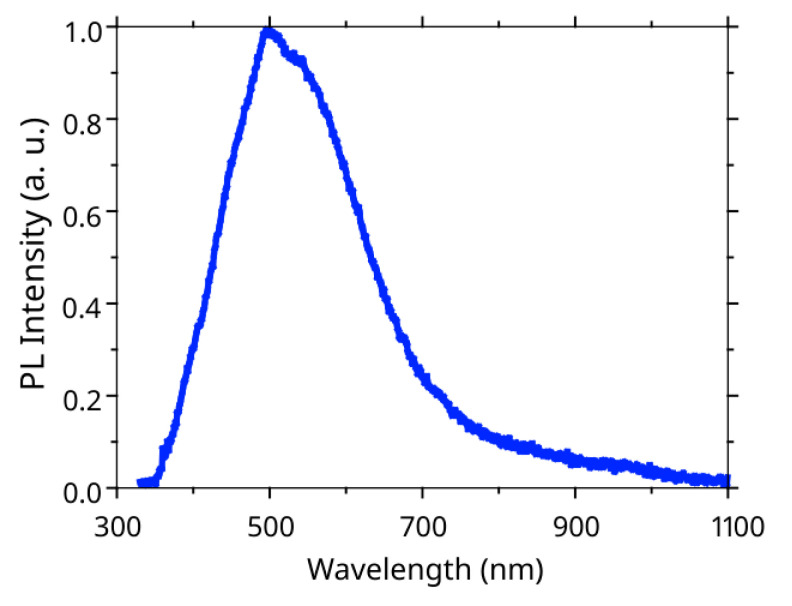
Normalized to the maximum PL response of the GaN/InGaN/GaN NWs synthesized on the Si/SiO_2_ substrate.

**Figure 4 nanomaterials-12-01993-f004:**
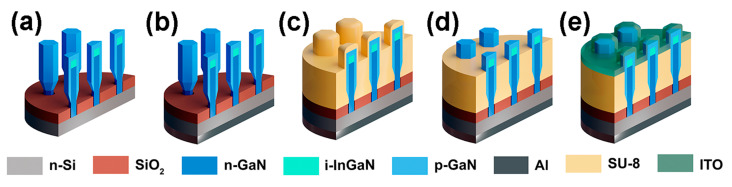
Workflow of NWs-based LEDs processing: (**a**) HF treatment, (**b**) evaporation of Al back contact, (**c**) covering with SU-8, (**d**) opening of NWs ends, (**e**) ITO sputtering.

**Figure 5 nanomaterials-12-01993-f005:**
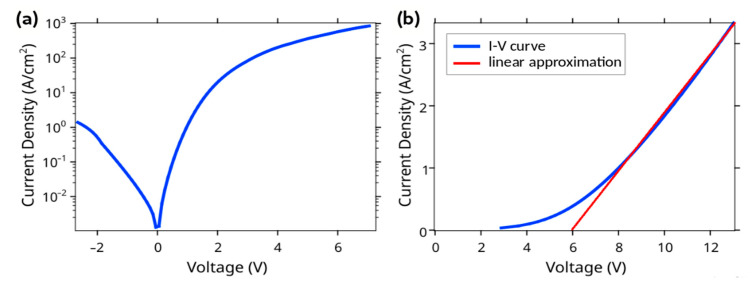
I-V characteristics of the (**a**) test n-GaN NWs-based structure and (**b**) n-GaN/i-InGaN/GaN NWs-based LED on Si.

**Figure 6 nanomaterials-12-01993-f006:**
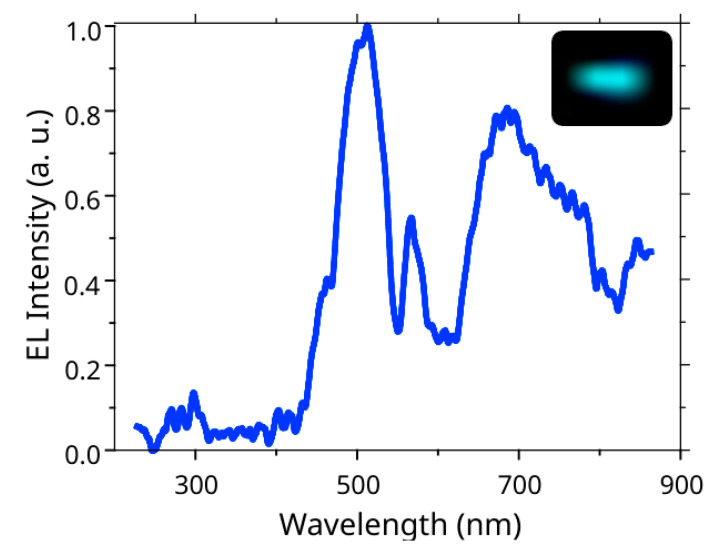
Normalized to the maximum EL spectrum of the n-GaN/i-InGaN/GaN LED device. The insert demonstrates an optical image of the operating NWs-based LED.

## Data Availability

The data presented in this study are available on request from the corresponding author. The data are not publicly available due to the author’s readiness to provide it on request.

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
