# Peer review of "Light-Emitting Diodes Based on InGaN/GaN Nanowires on Microsphere-Lithography-Patterned Si Substrates"

_nanomaterials, 2022, doi:10.3390/nano12121993_

Round 1

Reviewer 1 Report

Explain clearly the line of science, what has been done in this area, what need to be done and what you are trying to do in this paper

Editing of English language and style are required

Author Response

Comment 1. Explain clearly the line of science, what has been done in this area, what need to be done and what you are trying to do in this paper

Answer: As stated in the Introduction part of the manuscript, the development of InGaN/GaN NWs-based applications, especially LEDs, requires the regular arrangement of nanostructures. Time-consuming approaches for direct lithography, such as e-beam lithography [27] or focused ion beam milling [19,20], are not fully applicable for the prepartterning of large-area substrates.

Among other approaches, microsphere optical lithography, which provides sub-micrometer-scale lateral resolution, is one of the most versatile, scalable and cost -effective methods for photoresist patterning [30]. Moreover, the design of patterning can be easily tuned by the appropriate choice of diameter of microspheres, while spin-coating enables covering of the large-area substrates [31,32]. Selective-area epitaxy based on the patterning of Si/SiO2 growth substrates patterning with microsphere lithography allows the obtaining of the ordered arrays of NWs with narrow distribution in geometrical sizes, which is essential for device processing [33,34].

In this work we present microsphere lithography for Si/SiO2 substrates preparation, allowing further selective-area MBE growth of the regular arrays of n-GaN/i-InGaN/p-GaN heterostructures. We were able to fabricate NWs-based LEDs on Si substrates with several cm2 area. Moreover, the proposed approach for growth substrate patterning provides the handling of substrates several inches in diameter.

Comment 2. Editing of English language and style are required

Answer: The extensive English style and grammar editing was performed.

Reviewer 2 Report

The authors report an elaborate strategy to grow n-GaN/i-InGaN/p-GaN heterostructured nanowires arrays on Si/SiO2 substrates using molecular beam epitaxy methodology. This work turns out the one-of-a-kind pattern technique along with clear discussion on its mechanism. It provides a clever manner for implementing the self-induced disordered nanowire arrays for LED applications using MBE methodology. The authors demonstrate that this unique manner can achieve small footprint of wires as well as the applicable EL properties. I would say this work is insightful; however, it may be important to take care of some nuances associated with better details and for completeness required for this work to be published.

Comments

1. In terms of mass production, is it available to employ the MOCVD technique to achieve the selective epitaxy for heterostructured nanowire array fabrication similar to what you have done in this work? (Please show any possibilities in the Introduction part.)

2. In Page 1 (line 42), please place this abbreviation, “MBE”, at the end of molecular beam epitaxy (MBE).

Minor suggestion:

2. Can the authors show the quantum efficiency of these elaborate III-V heterostructured nanowires?

Author Response

Comment 1. In terms of mass production, is it available to employ the MOCVD technique to achieve the selective epitaxy for heterostructured nanowire array fabrication similar to what you have done in this work? (Please show any possibilities in the Introduction part.)

Answer: We thank the Reviewer for this valuable remark. Undoubtedly, MOCVD is the most promising epitaxial technique for III-V and III-N mass production, allowing the time-efficient synthesis of nanowire-based heterostructures simultaneously on the set of large-area substrates. MOCVD, similar to MBE, was employed to achieve the selective epitaxy of the ordered arrays of heterostructured nanowires [R1-3].

This discussion, as well as the corresponding references, were added to the manuscript.

  • Yuan, X.; Pan, D.; Zhou, Y.; Zhang, X.; Peng, K.; Zhao, B.; Deng, M.; He, J.; Tan, H.H.; Jagadish, C. Selective Area Epitaxy of III–V Nanostructure Arrays and Networks: Growth, Applications, and Future Directions. Applied Physics Reviews 2021, 8, 021302, doi:10.1063/5.0044706.
  • Nami, M.; Stricklin, I.E.; DaVico, K.M.; Mishkat-Ul-Masabih, S.; Rishinaramangalam, A.K.; Brueck, S.R.J.; Brener, I.; Feezell, D.F. Carrier Dynamics and Electro-Optical Characterization of High-Performance GaN/InGaN Core-Shell Nanowire Light-Emitting Diodes. Sci Rep 2018, 8, 501, doi:10.1038/s41598-017-18833-6.
  • Demontis, V.; Zannier, V.; Sorba, L.; Rossella, F. Surface Nano-Patterning for the Bottom-Up Growth of III-V Semiconductor Nanowire Ordered Arrays. Nanomaterials 2021, 11, 2079, doi:10.3390/nano11082079.

Comment 2. In Page 1 (line 42), please place this abbreviation, “MBE”, at the end of molecular beam epitaxy (MBE).

Answer: We thank the Reviewer for careful reading the manuscript. The correction was made.

Minor suggestion:

Comment 3. Can the authors show the quantum efficiency of these elaborate III-V heterostructured nanowires?

Answer: We thank the Reviewer for this comment. The studied NWs have a diameter of 200 nm, supporting the propagation of several optical modes. Radiation diagram of the array of LED NWs has a complex pattern with several peaks [R4-5]. For the correct estimation of the quantum efficiency of the fabricated devices, the optical setup with an integrating sphere should be employed. In our work we used a confocal optical scheme for EL measurements, that doesn’t allow the collection of all emitted light (due to low numerical aperture of the used objective). Therefore, we are not able to estimate the process efficiency.

  • Motohisa, J.; Kohashi, Y.; Maeda, S. Far-Field Emission Patterns of Nanowire Light-Emitting Diodes. Nano Lett. 2014, 14, 3653–3660, doi:10.1021/nl501438r.
  • Quan, L.N.; Kang, J.; Ning, C.-Z.; Yang, P. Nanowires for Photonics. Rev. 2019, 119, 9153–9169, doi:10.1021/acs.chemrev.9b00240.

Reviewer 3 Report

In the present manuscript, authors reported the molecular beam epitaxy of regular arrays n-GaN/i-InGaN/p- GaN heterostructured nanowires and tripods on Si/SiO2 substrates prepatterned with the use of cost-effective and rapid microsphere optical lithography. The authors experimentally showed that n-GaN NWs / n-Si interface demonstrated rectifying behavior and the fabricated n-GaN/i-InGaN/p-GaN NWs-based LEDs had electroluminescence in the broad spectral range with maximum near  500 nm, which can be employed for multicolor or white light screen development.

The manuscripts deserve to be published in Nanomaterials after addressing minor comments.

1.     It is recommended to use latest references in the introduction part of the manuscript.

2.     Figures 3, 5 and 6 need to be reformatted to match the standards of Nanomaterial journal.

3.     Check the manuscript for the grammatical errors and correct it.

4.     Cite the following reference.

https://www.sciencedirect.com/science/article/pii/S2211285521010028

Author Response

Comment 1.     It is recommended to use latest references in the introduction part of the manuscript.

Answer: We thank the Reviewer for this comment. We added the references to the latest works in the Introduction.

List of the added references.

  • Zhao, C.; Alfaraj, N.; Chandra Subedi, R.; Liang, J.W.; Alatawi, A.A.; Alhamoud, A.A.; Ebaid, M.; Alias, M.S.; Ng, T.K.; Ooi, B.S. III-Nitride Nanowires on Unconventional Substrates: From Materials to Optoelectronic Device Applications. Progress in Quantum Electronics 2018, 61, 1–31, doi:10.1016/j.pquantelec.2018.07.001.
  • Janjua, B.; Sun, H.; Zhao, C.; Anjum, D.H.; Priante, D.; Alhamoud, A.A.; Wu, F.; Li, X.; Albadri, A.M.; Alyamani, A.Y.; et al. Droop-Free Al x Ga 1-x N/Al y Ga 1-y N Quantum-Disks-in-Nanowires Ultraviolet LED Emitting at 337 Nm on Metal/Silicon Substrates. Opt. Express 2017, 25, 1381, doi:10.1364/OE.25.001381.
  • Janjua, B.; Sun, H.; Zhao, C.; Anjum, D.H.; Wu, F.; Alhamoud, A.A.; Li, X.; Albadri, A.M.; Alyamani, A.Y.; El-Desouki, M.M.; et al. Self-Planarized Quantum-Disks-in-Nanowires Ultraviolet-B Emitters Utilizing Pendeo-Epitaxy. Nanoscale 2017, 9, 7805–7813, doi:10.1039/C7NR00006E.
  • Alfaraj, N.; Mitra, S.; Wu, F.; Ajia, I.A.; Janjua, B.; Prabaswara, A.; Aljefri, R.A.; Sun, H.; Khee Ng, T.; Ooi, B.S.; et al. Photoinduced Entropy of InGaN/GaN p-i-n Double-Heterostructure Nanowires. Appl. Lett. 2017, 110, 161110, doi:10.1063/1.4981252.
  • Sadaf, S.M.; Zhao, S.; Wu, Y.; Ra, Y.-H.; Liu, X.; Vanka, S.; Mi, Z. An AlGaN Core–Shell Tunnel Junction Nanowire Light-Emitting Diode Operating in the Ultraviolet-C Band. Nano Lett. 2017, 17, 1212–1218, doi:10.1021/acs.nanolett.6b05002.

Comment 2.     Figures 3, 5 and 6 need to be reformatted to match the standards of Nanomaterial journal.

Answer: We thank the Reviewer for careful reading the manuscript. The corresponding figures (as well as Figure S1 in the Supplementary materials) were modified to meet the requirements of Nanomaterials journal.

Comment 3.     Check the manuscript for the grammatical errors and correct it.

Answer: The extensive English style and grammar editing was performed.

Comment 4.     Cite the following reference.

https://www.sciencedirect.com/science/article/pii/S2211285521010028

Answer: In accordance with the Reviewer comment this reference was cited in the Introduction (as Ref. 17).
